# Squeeze Behaviors of Magnetorheological Fluids under Different Compressive Speeds

**DOI:** 10.3390/ma16083109

**Published:** 2023-04-14

**Authors:** Hongyun Wang, Cheng Bi, Wenfei Liu, Fenfen Zhou

**Affiliations:** School of Intelligent Manufacture, Taizhou University, Taizhou 318000, China; emmawhy4@163.com (H.W.); liuwenfei45@163.com (W.L.); zhoufen8888fen@163.com (F.Z.)

**Keywords:** magnetorheological fluid, compression mode, viscoelastic material, Deborah number

## Abstract

The compression tests under the unidirection for magnetorheological (MR) fluids have been studied at different compressive speeds. The results indicated that curves of compressive stress under different compression speeds at the applied magnetic field of 0.15 T overlapped well and were shown to be an exponent of about 1 of the initial gap distance in the elastic deformation region and accorded well with the description of continuous media theory. The difference in compressive stress curves increases significantly with an increasing magnetic field. At this time, the continuous media theory description could not be accounted for the effect of compressive speed on the compression of MR fluid, which seems to deviate from the Deborah number prediction under the lower compressive speeds. An explanation based on the two-phase flow due to aggregations of particle chains resulting in much longer relaxation times at a lower compressive speed was proposed to explain this deviation. The results have guiding significance for the theoretical design and process parameter optimization for the squeeze-assisted MR devices such as MR dampers and MR clutches based on the compressive resistance.

## 1. Introduction

Magnetorheological (MR) fluids are smart materials that can transform from Newtonian fluid to Bingham fluid under an applied magnetic field [1,2]. They often exhibit viscoelastic properties in mechanics, which have properties of both viscous fluid and elastic solid [3]. The shear yield stress of MR fluids is an important argument in the applications, which determines the mechanical properties of MR appliance. The low shear-yield stress of MR fluids limits the application of MR fluids in the industry [4,5,6]. To obtain higher mechanical strength of MR fluids, new MR fluids with high magnetic performance have been invented [7,8]. Furthermore, more and more attention has also been paid to compressive mechanical properties in squeeze mode. The squeeze-strengthening effect of MR fluids was first proposed by Tang et al. [9]. At the same applied magnetic field, the mechanical performances of MR fluids are strongly dependent on the particle chain structure [1,10,11]. The arrangement of particles in MR fluids changes from disorder to order under the applied magnetic field. Furthermore, single chains are formed simultaneously, accompanying incomplete chains, such as broken, branched, isolated, etc. [10]. Compression can repair the weakness of chain structure in MR fluids and form a stronger column or body-centered cubic (BCC) structure, which improves the yield strength of MR fluids [11]. The structure evolution of MR fluids in squeeze mode has been validated by the computer simulation and examined by scanning electron microscopy [5,11].

In the study of the compressive properties of MR fluids, the influence factors, such as magnetic field (*B*), initial gap distance (*h*_0_), viscosity of carrier fluid (*η*), and particle volumes concentration (*φ*), have been widely discussed [12,13,14,15,16,17,18,19,20]. Researchers found that the compressive force (*F*) mainly depends on the power law relationship of *B* as *F*∝*B^a^* (*a* is about 2) [12,13,14]. Guo et al. studied the relationship between *F* and the instantaneous gap distance (*h*) or compressive strain (*ε*) of MR fluids as *F*∝*h^n^* or *F*∝(1-*ε*)*^n,^* and the exponent *n* is about −3 to −2 [15]. Vicente et al. and Ruiz-L’opez et al. reported that the compressive stress (*P*) increases proportionally with *φ* as *P*∝*φ* [16,17]. They proved that higher *φ* help the particles in MR fluids form a more robust structure in a high-concentration suspension. *F* was also discovered to increase with increasing *η* under the same *h*_0_, which was attributed to the fact that the high-viscosity carrier fluid can maintain the particle structure more effectively [15,17]. Recently we have studied the compressive properties of MR fluids under different magnetic fields and different initial gap distances and compared the measured and calculated compressive stresses [18]. Tang et al. have proposed that the nominal shear-yield stress (*τ_E_*) is in direct proportion to *P* as *τ_E_*∝*P* [11]*. τ_E_* of MR fluids was found to be about ten times higher under compression than under shear at the same *B* [10,18,19,20]. Carlson has found that *τ_E_* of MR fluids is proportional to *φ* with an exponent of 1.52 (under high *φ*), that is *τ_E_* ∝*φ*^1.52^ [21]. Guo et al. found that *τ_E_* has a relationship with *h* as *τ_E_*∝*h^−k^*, and *k* is a relevant parameter with the squeeze-strengthen effect [15]. Luo et al. studied the main factors influencing the quasistatic compressive properties of MR fluid. At the same time, the initial stress (*σ*_0_) and the compressive yield stress (*σ*_y_) were found to depend on the power law relationship of *B* or *φ*, including *σ*_0_∝*φB^2^*, *σ*_y_∝*B^2^* and (*σ*_y_-*σ*_0_)∝*φ^2^* [22]. P. Pei and Y. B. Peng conducted a systematic molecular dynamics simulation for squeezed MR fluids. They demonstrated that the squeeze-strengthening effect of MR fluids is improved with increasing *B*, but weakened with increasing squeeze rate at more than the critical squeeze rate [23]. Horak has developed a new nonlinear model of MR fluids under quasistatic squeeze flow based on the theories of continuum mechanics and tensor transformation [24]. Wang et al. have investigated the fracture mechanism of chains of MR fluids and established the fracture criterion based on the magnetic force model [25]. They found that the interaction between hydraulic force and magnetic force leads to fracture. Zheng et al. have researched the relationships among *h*_0_, *B*, *φ* and the sealing properties of MR fluids on the seal of MR fluids based on the quasistatic tensile tests and proposed a theoretical model to calculate the burst pressure of MR fluid seal according to the theory of magnetic dipole [26].

In recent years, the squeezing–strengthening effect of MR fluids has been used to guide the scheme design of ultra-strong MR devices. Singh et al. designed a compact squeeze film MR damper to decrease vibrations at different working conditions [27]. Ruan et al. exploited a new MR damper under the squeeze-valve bi-mode with a maximum damping force of about 6.5 kN, which is very potential in the application of the semi-active control [28]. Wang et al. used the Navier-Stokes equation to simulate the 2D squeeze flow in the squeezed MR damper and obtained the 2D speed field and distribution of pressure of MR fluid [29]. Wang et al. developed an MR brake under compression-shear mode, and the maximum torque produced by this brake is 241 Nm, which is about 17.9 times that of a non-compression brake [6]. Huang et al. proposed a heavy-duty MR mount with a flow and squeeze model, providing a maximum damping force of 18.843 kN [30].

For the design and control of squeezed MR devices, the compressive speed (*v*) is an important design parameter. However, much attention has been paid to *v* of MR fluids under squeeze mode by many researchers. Mazlan et al. reported that the effect of *v* on the stress-strain curves of MR fluids is negligible under the compressive speed of 17–170 μm/s [31]. Guo et al. observed that when the particle concentration of MR fluid is 15%, *v* of 8.3–33.3 μm/s has very little impact on the force-gap relationship [32], which agrees with the result of Mazlan et al. [31], but *F* increased with decreasing *v* at the 30% MR fluid. They thought that stronger structures could be formed at slower *v* for the high-concentration MR fluid. To MR homologue-ER fluid, Tian et al. at *v* of 2.6–52 μm/s [33] and McIntyre and Filisko at *v* of 2.4–38.5 μm/s [34] obtained that *F*/*P* of ER fluid decreases with increasing *v*. These different results showed that the influence of *v* on the MR behavior is different under different conditions. However, the effect of the compressive speed on the compressive performances of MR fluids in squeeze mode has not been thoroughly studied.

In this study, the quasistatic unidirectional compressions of MR fluids in a series of different compressive speeds and magnetic fields have been investigated in the constant area. The results showed that compression at a lower compressive speed could make MR fluids have stronger compressive/shear yield resistance. The results are opposite of what we conventionally expect for the low Deborah number, which is a rheological parameter characterizing the viscoelasticity of fluid, that a low Deborah number generated by a low compressive speed corresponds to a liquid-like mechanics performance. The results were attributed to the sealing effect and the squeeze-strengthening effect caused by robust chain structures during compression at the low compressive speed.

## 2. Experiments

The apparatus of compressive experiment employed in this study is the MCR 302 rheometer of Anton Paar, as shown in Figure 1a. Figure 1b shows the schematic diagram of the rheometer. The radii *R* of the plates are both 10 mm. The maximum-measured normal force *F* is 50 N. The compressive stress *P* was represented as *P* = *P*_1_ − *P*_2_ during compression, and it could be calculated by *P* = 2*F/*π*R*^2^ [35]. The compressive strain *ε* can be described as *ε =* (*h_0_ − h*)/*h_0_*. The magnetic field of the rheometer was in the range of *B* = 0–1 T, corresponding to the applied current of 0~5 A. Figure 1c,d indicate the magnetic induction line and the magnetic flux density distribution of the MCR 302 rheometer at the applied current of 3.0 A, respectively. The MR fluid (MRF-2035) produced by Ningbo Shangong Co., Ltd., Ningbo, China is with *φ* = 35%.

The gap distance between the two plates was first adjusted to *h_0_* = 1 mm. After applying a current to the MR fluid for about 40 s, compressions were performed. The different applied currents of 0.41 A, 1.15 A, 2 A, and 2.88 A correspond to different applied magnetic fields of 0.15 T, 0.3 T, 0.45 T, and 0.6 T, respectively. Under different magnetic fields, seven compressive speeds of 1, 5, 10, 25, 50, 75, and 100 μm/s were applied, respectively. The applied current remained constant during compression and turned off after the compression. Because of the force sensor deformation, the actual displacement under a compressive force was smaller than the nominal displacement. So, the actual compressive speed is less than the nominal one. During the experiment, the compressive speed could be flexibly controlled by rising and falling in a linear/logarithmic way. All experiments were performed at a room temperature of 25 °C.

## 3. Theoretical Analysis

The relationship of shear stress versus the shear rate of MR fluids has been measured by a MCR 301 rheometer at *h_0_* = 1 mm when the applied currents changed from 1 to 5 A. It showed that the static shear-yield stresses, which were obtained from the ramp-up curves of shear stress versus shear rate by the rheometer, depend on the power law relationship of *B*, as shown in Figure 2:(1)$$\tau_{\mathrm{Esta}}=10226B^{1.53}$$

This result is in agreement with the previous investigation [16,32].

According to the Bingham constitutive equation and the lubrication approximation under constant area, the compressive stress *P* can be described as [36]:(2)$$P=\frac{2r_{0}}{3h}\tau_{E}+\frac{4r_{0}\tau_{E}}{7h}\sqrt{2S}=\frac{2r_{0}\tau_{E}}{h}\left(\frac{1}{3}+\frac{2}{7}\sqrt{2S}\right)$$
where *r_0_* is the radius of the upper plate, *τ_E_* is the shear yield stress, and *S* is the plasticity number as *S* = *ηRv/h*^2^*τ_E_* (*v* is the compressive speed and in the Bi-viscous model, *η* is the pre-yield viscosity). In this experiment, taking *η =* 0.24 Pa·s, *R* = 10 mm and substituting Equation (1) into Equation (2), the compressive stress can be calculated. Meng and Filisko found that the compressive stress will predominate if *S* < 0.5, and the second term in Equation (2) has little contribution to the total compressive resistance, and it may be negligible [16,34]. We try to seek the maximum value of *S* when *B*_min_ = 0.15 T, *v*_max_ = 100 μm/s, *h*_min_ = 0.5 mm (the units of *τ_E_*, *B*, and *h*, are Pa, mT, and m, respectively). Under this condition, the calculated *S*_max_ was 0.43. Therefore, Equation (2) can be simplified as
(3)$$P=\frac{2R}{3h}\tau_{E}$$

Guo et al. have demonstrated that *τ_E_* of MR fluids has an exponential relationship with *h* about [15]:(4)$$\tau_{E}=\frac{K}{h^{n}}$$
where *K* is constantly relative to the material of MR fluids, and *n* is relative to the squeeze-strengthen effect and the sealing effect. Substituting Equation (4) into Equation (3), Equation (3) can be expressed as
(5)$$P=\frac{2RK}{3h^{n+1}}$$

For the maximum instantaneous gap distance *h*_max_, the compressive stress *P_c_* can be represented as
(6)$$P_{c}=\frac{2RK}{3h_{\max}^{n+1}}$$

For a given MR fluid, *n* is a material constant that is considered not to vary with *h* in the range of considered gap distance. Dividing Equation (5) by Equation (6), the normalized compressive stress *P/P*_c_ is represented by the normalized instantaneous gap distance *h/h*_max_ as
(7)$$\frac{P}{P_{c}}={\left(\frac{h_{\max}}{h}\right)}^{n+1}$$

Taking *h*_max_ = 1 mm, the natural logarithmic form of Equation (7) becomes
(8)$$\ln P=\ln P_{c}-(n+1)\ln h$$

According to Equation (6), *P_c_* is a constant when *h*_max_ = 1 mm. *n* should be a constant to a given MR fluid. Equation (8) means that in ln–ln plot, the intercept is ln *P_c,_* and the slope is (*n +* 1). Moreover, ln *P* shows linear change with ln *h*. So, the magnetic field, the compressive speed, and the material factor *K* should affect neither *n* nor ln *P_c_* according to Equation (8).

## 4. Results and Discussions

Figure 3 shows the curves of *P* versus *h* of compressions when *B* was applied for 0.15 T, 0.3 T, 0.45 T, and 0.6 T under different compressive speeds *v*. Seven curves show a roughly similar trend, and *P* increases quickly with the decreasing *h*. When *B* = 0.15 T, *P* increases quickly with the decrease in *h*, as shown in Figure 3a. The compression of MR fluid can be divided into two processes. In Ⅰ process, *P* increased almost linearly with the decreasing *h*. *ε* is no more than 0.012 in this process, which can be called the elastic deformation process. In the Ⅱ process, *P* increased slowly with decreasing *h*, which can be called the plastic flow process. The curves at different *v* overlap for the most part. The overlapped curves can be fitted with an exponent of 1.06. The viscous stresses under a zero magnetic field are shown in Figure 3a. Experimental verification clearly shows that viscous stress has little contribution to the total compressive stress. The result for *v* effect on *P* is consistent with that by Mazlan et al. [31], observing that *v* has little effect on *P*.

When *B* = 0.3 T, as shown in Figure 3b, the compressive curves most coincide with each other as *v* decreasing from 100 to 10 μm/s under the compressions. With the further decrease in applied *v* to 5 μm/s, the compressive curves are almost consistent at large *h,* and there is a small deviation when compressions are over. With decreasing *v*, this deviation is more obvious. A similar result has been obtained for MR fluids under *B* = 0.28 T [32].

When *B* = 0.45 T, as shown in Figure 3c, all curves are no longer well-overlapped at the end of the compressions, and the difference between curves becomes more remarkable. *P* at a low *v* is higher than that at a higher *v* under a certain *h*. Four compressive curves with compressive speeds from 100 to 25 μm/s only indicate a small deviation when the compressions are over. With a further decrease in *v*, the deviation is more significant among the curves. In Figure 3d, all seven curves when *B* = 0.6 T show obvious differences among each other, which is contrary to the results of Guo et al. [15], showing that the normal forces increase faster at higher *v* under *B* = 0.43 T. However, it is consistent with the experimental results for ER fluids [33,34].

Comparisons between two different *v* (1 and 100 μm/s) under four different *B* (0.15 T, 0.3 T, 0.45 T, and 0.6 T), respectively, have been carried out during compression, as shown in Figure 4. It indicates that *P* curves overlap well at low *B,* and *v* has little effect on *P* under these conditions. With the further increase in *B*, *P* is no longer overlapped, and the differences among *P* curves increase with decreasing *v*.

According to the experimental values in Figure 3, the calculated values of logarithm for *P* and *h* at *B* = 0.15 T, *B* = 0.3 T, *B* = 0.45 T, and *B* = 0.6 T are displayed in Figure 5a–d, respectively. The curves at different *v* can be fitted by linear functions. When *B* = 0.15 T, the curves applied at different *v* overlap over most of the ranges. In the plastic flow process, a linear line with a slope of −2 is well-used to capture ln*P* with ln*h* at different *v*. The fitted curve is also shown in Figure 5a. Based on the continuum media theory, the power law relationship between *F*/*P* and 1/*h* with the index *n* = 1 is predicted [15]. Therefore, the value of the theoretical slope (*n +* 1) should be 2. It is an obviously weak gap distance dependence under this low field of 0.15 T. Therefore, we may conclude that *n* should be about 1 at this low *B* in Equation (4). Figure 5a indicates that the principle of the continuous media theory may well depict the squeezing behavior of MR fluid at this low *B*.

When *B* = 0.3 T, the curves at different *v* overlap for the most part except for that of *v* = 1 μm/s and *v* = 5 μm/s, as shown in Figure 5b. As shown, a good linear relationship is theoretically predicted with a slope of 2 in the plastic flow process for *v* from 100 to 10 μm/s. Additionally, the curve of *v* = 1 μm/s fitted with a slope of −2.43 is shown in Figure 5b.

When *B* = 0.45 T, the slopes are 2.04 (100 μm/s), 2.06 (75 μm/s), 2.14 (50 μm/s), 2.34 (25 μm/s), 2.85 (10 μm/s), 4.11 (5 μm/s), and 7.46 (1 μm/s), as shown in Figure 5c. According to Equation (8), it seems that the slopes are dependent on neither *B* nor *v*. However, Figure 5c shows that the experimental slopes are no longer in agreement with the theoretical value, and they are different from each other. It increases with the decrease in *v*. An average value of the slopes is 3.29.

With the further increase in *B* to 0.63 T, the difference between the slopes becomes more pronounced, as shown in Figure 5d. The slope range is 2.33–16.30, and the mean value is 5.79. It also displays a trend that the slope decreases with the increase in *v*. The linear function for the curve at *v* = 1 μm/s is ln*P* = 4.92–9.52 ln*h*. *P* depends more strongly on *v* than that at *v* = 100 μm/s given by ln*P* = 4.86–2.42 ln*h.*
Figure 5c,d show that the deviations are accelerated with decreasing *v* from 0.45 T to 0.63 T. All experimental results of the slope greater than 2 imply that *P* of MR fluids is underestimated by the continuum media theory, especially under high *B* and low *v*. Additionally, *P* increases with decreasing *v* more quickly than the theoretical prediction under the same initial gap distance. From Figure 5c,d, we can see that *v* and *B* affect the intercept ln*P_c_*. ln*P_c_* range from about 4.81 to 8.88 at *B* = 0.45 T with an average value of 4.85 and from about 4.87 to 5.02 at *B* = 0.63 T with an average value of 4.96 under different *v*, respectively. It means that *v* and *B* have little effect on ln*P_c_.* Overall, the values of the experimental slopes are all much larger than the theoretical values of 2 at a higher *B*. It shows that the description based on the continuous media theory can not account for the effect of *v* on the squeezed MR fluids.

According to Equation (3), the shear-yield stress under the compression can be calculated under different *B* and *v*, which is often called the normal shear-yield stress *τ_Enor_*. The comparison between *τ_Enor_* and the static shear-yield stress *τ_Esta_* at different *v* under *h* = 0.91 mm and *h* = 0.81 mm are shown in Figure 6a and Figure 6b, respectively. It indicated that *τ_Enor_* does increase with increasing *B*, which is good in agreement with Ruiz-López et al. [37], and decreasing *v*. *v* has little effect on *τ_Enor_* at the lower *B* and has a great effect on *τ_Enor_* at the higher *B*, as shown in Figure 6. *τ_Enor_* under the compression is obviously higher than *τ_Esta_* without compression. In addition, the range of the pressure sensor used in the dynamic compression is only 50 N. The maximum compressive stress can only reach about 300 kPa, so only part of *τ_Enor_* is shown in Figure 6b.

MR fluids are often considered viscoelastic material under high *B* [16,21]. The yield stress of MR fluids is usually used to indicate the degree of solidification under *B*. Deborah number (*De*), which is a dimensionless quantity in rheology, is applied to represent the fluidity of materials at specific conditions. *De* is an important parameter to evaluate the viscoelasticity of fluid [38]. It can be defined as the ratio of relaxation time (*θ_r_*) to observation time/testing time (*θ_t_*) of the material mechanical response at observing conditions [39]. *De* may be represented as
(9)$$De=\frac{\theta_{r}}{\theta_{t}}$$

Equation (9) shows that when the value of *De* is smaller, the material shows Newtonian fluid behavior. Additionally, when the value of *De* is larger, it shows non-Newtonian solid-like behavior. Testing time can be described as [39]
(10)$$\theta_{t}=\frac{h_{0}^{2}}{v}$$
where *h*_0_ is the initial gap distance, and *v* is the test speed or compressive speed. Substituting Equation (10) into (9), *De* can be rewritten as
(11)$$De=\frac{v\theta_{r}}{h_{0}^{2}}$$

According to Equation (11), a low *v* with a long *θ_t_* corresponding to a small *De* results in a liquid-like mechanical property of materials, and a high *v* with a short *θ_t_* corresponding to a large *De* results in a solid-like mechanical property. Therefore, we can expect that the compressive/shear-yield stress of MR fluid will increase when *v* increases, and the MR fluid shows a solid-like property. However, the results showed in Figure 3 and Figure 6 that a low compressive speed corresponds to a large compressive/shear-yield stress are contrasted to the result predicted by *De*. So, the obtained experimental results of MR fluid and the prediction of *De* in the viscoelastic material seem to have deviation under lower test speeds. There must be some reason for this opposite result.

The strength of MR fluids is generally considered to be determined by the applied *B* magnitude. Except for *B*, the chain microstructures of MR fluids under *B* have a great effect on the mechanical property of MR fluids [1,9,10,11]. The microscopic mechanism of the MR effect under *B* is shown in Figure 7. As shown in Figure 7a, the particles are randomly distributed under a zero *B*. Magnetized particles form chains, including broken, branched, and isolated incomplete chains in the direction of *B*, as shown in Figure 7b. As shown in Figure 7c, more complete chains will be formed under *B*. When the MR fluid between two parallel plates is sheared, its volume remains constant. However, the volume of the MR fluid will be reduced with a decreasing gap in the constant area compression. The base liquid and the partially magnetized particle will be squeezed out. Under extreme conditions, this reduction of volume is due to only being out of the base fluid. The magnetized particles are assumed to rearrange themselves in the direction of *B*, which prevents the flow of liquid. This is the so-called sealing effect [15,16]. It is assumed that the magnetized particles and the base liquid in the MR fluid exist in relative motion during compression [16,40]. The conclusion that a lower compressive speed resulted in a higher and stronger structure of the interconnected particle chains and a greater resistance preventing the movement of the base liquid can be obtained. At the same time, the particle volume fraction will be increased as soon as the sealing effect happens, which will cause an increase in the yield strength because the yield stress of MR fluids is proportional to the volume fraction of the particle [19,21]. As analyzed above, the notable improvement of the chain structure inter-connectivity at the low *v* and the increasing volume fraction lead to the increase of the compressive/shear-yield stress.

In addition to the sealing effect, the squeeze-strengthening effect is another important factor that affects the variation of yield stress during compression. Tao found that the chain ends are the weakness of the MR microstructure because of the wall effect [10]. Compression can repair the weakness of the chain structure, which leads to the formation of more robust columns/body-centered cubic (BCC) structures [10,11,18]. Columns/BCC structures will become shorter and thicker during compression, which results in the cross-linking of particle chains. All of these greatly increase the chain structural strength and compressive/shear strength. Additionally, the compressed process of MR fluids is a dynamic-forming process in which the old chain breaks and the new chain forms. As is known to all, the magnetizing time of MR particles and the formation of weak single-chain structures are in the millisecond order, but the formation of the robust columns/BCC structures is generally on the order of seconds. Figure 3c,d and Figure 6 show that the lower the compression speed, the higher the compressive/shear-yield stress. That means that the formation rate of the new chain is higher than the breaking rate of old chains under the low compressive speed, and the robust columns/BCC structures are more likely to be formed at low compressive speeds. This is in agreement with the result for ER fluids [33,34], which showed that the particle chain structures of ER fluid are damaged more seriously when the compressive speed is high, resulting in lower current density and lower compression stress. Li et al., studying the shear performance of MR fluid, also found a similar phenomenon in that the shear stress increases first and then decreases with an increasing shear rate in a certain range [40]. They owed to the effect of shear rate on the destruction and reconstruction of particle chain structure in MR fluids. Therefore, the most fundamental reason causing the difference in yield strength of MR fluids is the effect of compressive speed on the collapse and reconstruction of the particle chain structure.

The MR fluid behaves as fluid-like under the zero *B* and as solid-like under the high *B*. Therefore, it is often used as a viscoelastic material. The particles of the MR fluid are magnetized under the applied *B*, and the robust columns/BCC structures are generated along the direction of *B* during compression. The squeeze flow of MR fluids is often regarded as a two-phase flow of base liquids and particles under the magnetic field. A narrow channel will be formed between the chain structures. During compression, the base liquids will squeeze the chain structure when passing through this narrow channel, and the particle of the chain will be dragged, which can be called the viscous force *F_D_*. *F_D_* will destroy the chain structure. According to *F_D_* = 3*πηdv_p_*, a lower *v* corresponds to a lower particle speed *v_p_* and a weak *F_D_*. A weak *F_D_* at a low *v* will cause a weak destructive force and a strong interaction between particles. At the same time, the chains tend to re-establish themself during compression because of the attraction between magnetized particles, which can be called the attracting force *F_A_*. Thus, the combined action of *F_D_* and *F_A_* controls the final chain structure of the particles during compression. At the same *B* and compressive force, *F_D_* will decrease with decreasing *v*, so *F_A_* should increase. It means that MR fluids show a more solid-like mechanical property under a lower *v*. However, it seems to be contrary to the viscoelastic properties of the fluid that is characterized by *De*. According to Equation (11), *De* should decrease with the decreasing *v* to the most viscoelastic material. However, in MR fluids, there are the attracting forces *F_A_* between magnetized particles; the low *v* results in more robust chain structures. The relaxation time *θ_r_* of robust chain structures is greatly increased as well. As shown in Figure 3c,d and Figure 6, the increase of *θ_r_* should be greater than the decrease of *v* under the lower *v*, which leads to the high value of *De* and the solid-like property. Therefore, we can conclude that more robust chain structures at a lower *v* result in a larger *θ_r_*, which plays a decisive role in a high *De*.

MR fluids often work in three basic geometrical arrangements: flow mode, shear mode, and squeeze mode. Among these modes of MR fluid, the geometrical arrangement designed for the squeeze mode usually generates the highest resistance [19]. Many applications of MR fluids under squeeze-assisted mode have been suggested. The high-performance damper has always been in great demand in civil engineering. Ruan et al. developed an MR damper with squeeze-valve bi-mode that has been proven to have great potential application in semi-active control fields [28]. Additionally, the floor vibration and the vibration isolation of heavy-duty machines have great requirements for designing a vibration mount with a large force output. Huang et al. introduced a heavy-duty MR mount under a flow-squeeze model that has employed a novel squeeze-flow-valve to come into two symmetric radial flow channels, which can improve the *B* [30]. Sarkar et al. developed an MR brake under compression-shear mode that can generate larger torque than that operating only under shear [41]. Compressive frequency and compressive amplitude are important parameters in the design of MR devices with squeeze-valve mode. When the compressive amplitude is the same, *v* increases with increasing compressive frequency. Similarly, *v* increases with increasing compressive amplitude when the compressive frequency is the same. Compressive frequency or compressive amplitude affects the damping force by changing *v*. Therefore, the design and control of MR devices based on the compressive resistance of MR fluids will fail if the *v* influence is not considered. In further research, it is necessary to focus on the *v* optimum for practical squeezed MR devices.

## 5. Conclusions

MR fluids are often regarded as non-Newtonian fluids under *B*. Researchers usually use non-Newtonian continuum media theory to represent the response of a fluid under unidirectional compression. In this study, the theory for quasistatic compression was extended by normalized logarithm form. Compressions of MR fluids were investigated at different *v* using an MCR 302 rheometer. Experimental results of compression show that high *P* could be obtained in the case of the high *B* and the low *v*., and the theory underestimates the compressive resistance at a low *v* and a high *B*. The shear-yield stress of squeezed MR fluids is obviously higher than that of MR fluids without being squeezed. A lower *v* leads to a higher compressive/shear-yield stress under a high *B*. The effect of *v* on the compression of MR fluids is opposite to the *De* prediction at a lower *v*. Under slow compression, the aggregation of the chain structure leading to a longer relaxation time is currently used to account for this deviation.

## Figures and Tables

**Figure 1 materials-16-03109-f001:**
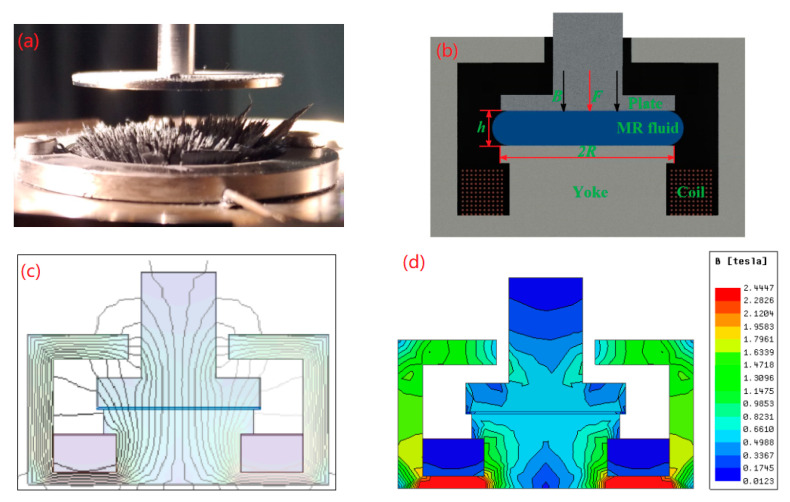
MR experimental setup. (**a**) MR fluid at applied magnetic field for MCR 302 rheometer; (**b**) Schematic of the test system; (**c**) Magnetic induction lines of the rheometer under 3A current; (**d**) Magnetic flux density distribution of the rheometer under 3A current.

**Figure 2 materials-16-03109-f002:**
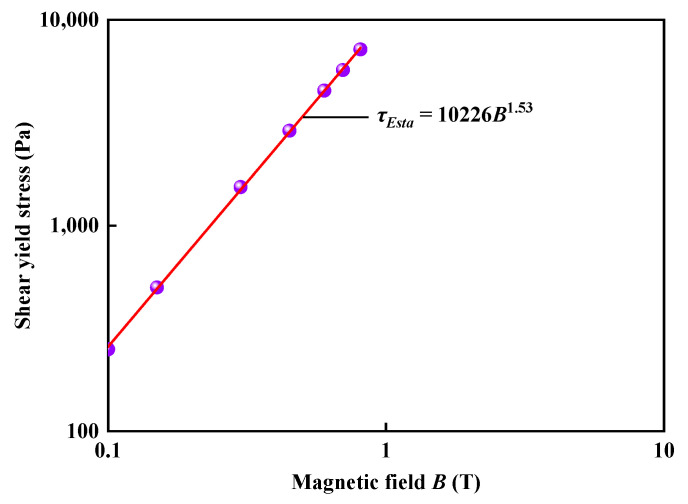
The shear yield stress of the MR fluid versus magnetic fields.

**Figure 3 materials-16-03109-f003:**
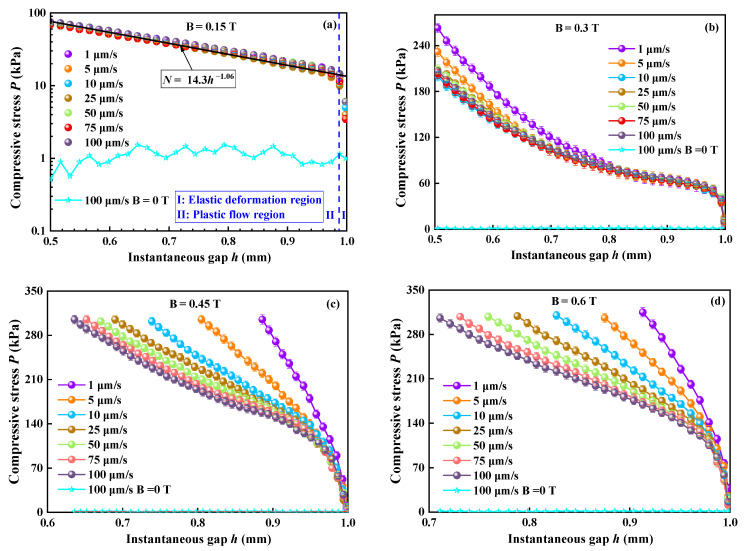
The curves of compressive stress under different compressive speeds. (**a**) *B* = 0.15 T; (**b**) *B* = 0.3 T; (**c**) *B* = 0.45 T; (**d**) *B* = 0.6 T.

**Figure 4 materials-16-03109-f004:**
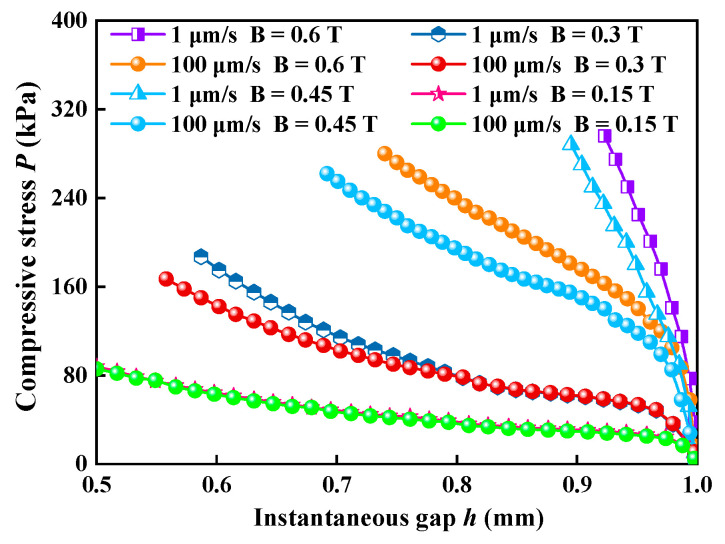
Comparison of the value of compressive stresses under two different compressive speeds (5 and 10 μm/s) and different magnetic flux densities (0.15 T, 0.3 T, 0.45 T, and 0.63 T).

**Figure 5 materials-16-03109-f005:**
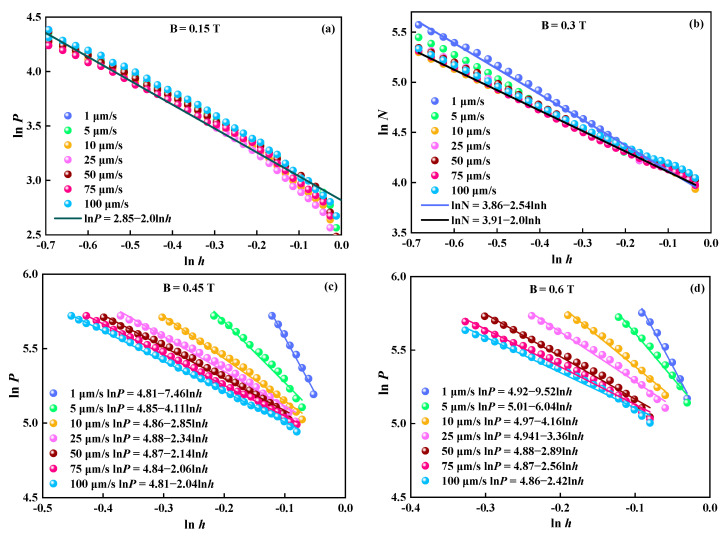
Logarithmic curves of compressive stress with different compressive speeds. (**a**) *B* = 0.15 T; (**b**) *B* = 0.3 T; (**c**) *B* = 0.45 T; (**d**) *B* = 0.6 T.

**Figure 6 materials-16-03109-f006:**
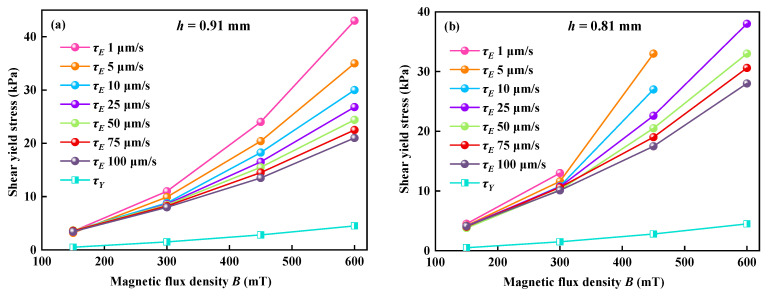
Comparison between normal shear-yield stresses yield and static shear-yield stresses under different compressive speeds and instantaneous gap distances. (**a**) *h* = 0.91 mm; (**b**) *h* = 0.81 mm.

**Figure 7 materials-16-03109-f007:**
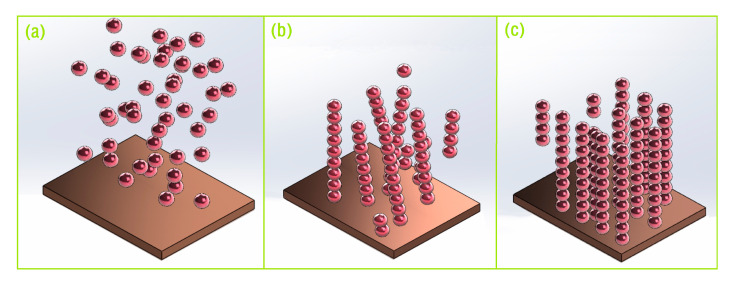
The schematic view of the formation of MR fluid microstructure: (**a**) under zero magnetic field. (**b**) under an applied middle magnetic field. (**c**) under an applied high magnetic field.

## Data Availability

Not applicable.

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
