# Peer review of "Squeeze Behaviors of Magnetorheological Fluids under Different Compressive Speeds"

_materials, 2023, doi:10.3390/ma16083109_

Round 1
Reviewer 1 Report
Referee Report
on paper “Squeeze behaviors of magnetorheological fluids at different compressive speeds and magnetic fields” (materials-2285909) by authors Hongyun Wang, Cheng Bi, Wenfei Liu and Fenfen Zhou submitted to Materials
This is interesting paper. It reports about the correlation between the external conditions (different compressive speeds and magnetic fields) and the behavior of the MR fluids. There are several methods and techniques were used for investigations and characterization. The paper can be interesting for specialists in the field of magnetic fluids for magnetorheological polishing and other functional applications . The obtained experimental results are interesting and reliable. However, paper needs some improvement only after which it can be accepted. At this stage, my decision is minor revision. But I hope that after brutal revision this paper can be accepted in Materials. I feel good potential for accept revised paper.
1. I feel that Abstract can be revised. Now it seems quite general. Authors no need describe the processes (what was done during investigations), they must highlight most important results with brief explanation the nature of the effect (highlight what was observed).
2. In Abstract please highlight the practical importance of the results and the possible applications.
3. Please compare your results with previous studies and mention clearly how your work is important in comparison to already been reported. I feel that Introduction seems quite poor. The choice of the research object is attractive. Please discuss the most promising magnetic materials for MR fluids. Not only Fe particles dispersed in polymer matrix attract great attention. There are several complex oxides can be successfully used for magnetic liquids (10.1016/j.jallcom.2018.04.150; 10.1016/j.jmmm.2015.05.076).
4. Authors claimed that “…magnetic flux density generated….was in the range of 0-1 T…” (p.3 line 120). But Tesla is unit for magnetic induction field. Magnetic flux density unit is weber (Wb) in SI and maxwell in CGS. It will be better say “…magnetic field generated…was in the range of B=0-1 T…”. The same comment fo title of the X-axis on Figure 2.
5. There are some typos and grammatical errors in the text. Please check this and revise this.
6. My decision is minor revision. I feel that after brutal revision it can be considered again.
Reviewer 2 Report
1. The references cited in this work are too old. There are many articles on the compressive (or squeeze) properties of MRF published since 2020. There are several paper where you can find similar result to those presented in this work.
2. The technical novelty and contribution is triavial. The results presented in this work can be easily expected without testing. For example, the higher compressive yield stress at a lower compressive speed under the same magnetic field.
3. In the compressive mode, how to define compressive yield stress or shear yield stress? Why is the unit of compressive stress N? Something is wrong.
4. The magnetic field analysis of the apparatus shown in Figure 1 should be provided to see the field direction.
5. The distance h in Figure 1 is originally constant. Isn't it? The change of h depends on compressive force, compressive speed and magnetic field intensity. How to define the instantaneous h and how to measure it ? The relationship of these parameters needs to be presented.
6. What kind of applications can you apply the results achieved in this work? please give a couple of examples and operating principle of the applications.
7. It is well known that there is a short time for reversibility between fluid and solid phase. How to measure the yield stress with a fast velocity?
Reviewer 3 Report
This paper investigated the quasi-static unidirectional compression of MR fluids, in different compressive speeds.
My main concern about this paper is its similarity. By using Turnitin, I have found 27% similarity of this research againts the previous study. One important paper by the authors published in Materials 2022, is "A Comparative Analysis of Measured and Calculated Compressive Stresses of Magnetorheological Fluids under Unidirectional Compression and Constant Area". this paper deals with quite similar topics. However, this paper was not critically reviewed in the present study.
Moreover, I suggest the authors to clearly state about the new contributions of this work in the introduction part.
Reviewer 4 Report
Research on the squeeze-mode characteristics of MR fluids has been conducted for a long time, and many research papers have been reported.
Followings are comments for this paper.
1. The originality and technical contribution to the related engineering field need to be clearly described in the paper. Compared to previous research papers, it is necessary to explain in detail what is novel about this paper.
2. It is necessary to explain in detail how the results of this paper can be used in MR fluid application research.
Reviewer 5 Report
The authors studied the theory of quasi-static compression of magnetorheological fluids. It is shown that high compressive stress can be obtained under a strong magnetic field and low compression speed. Aggregation of the chain structure occurs under slow pressure. The phenomena are explained on the basis of the two-phase flow model and the change in the relaxation time when the flow rate changes.
*The work is written clearly and distinctly.
*Graphs and formulas are clear and informative.
*The work can be published in the presented version, after minor corrections.
Notes for correction:
unclear records:
written ‒ ”….gap h/ compressive strain ε……?” ……………… … 47
written ‒ ”…τE ~φ1.52….?” ………… …. 59
written ‒ Fig or Figs ? ”….Figs. 5 (c) and (d)…”… 265 “….Figs. 6 (a) and (b)……” ………… 275
written ‒ …”….τY …”………… 279 or “….τY…”.. ? ……………………… .397
Conclusion:
The work can be published in the presented version, after minor corrections.
Round 2
Reviewer 2 Report
The revised version is not satisfied in its current form.
1. The authors should submit the separate response to the reviewer’s comments and the modified parts need to be denoted by different color. Otherwise, it is very difficult to see the modified part of each comment.
2. This reviewer has found that many parts of this paper are similar to the published articles as below (blue color). I found some figures are almost same as the previous ones, but presented by changing the unit of x-axis or y-axis. The authors should clearly address on the technical difference of this paper compared to the previous works.
3. Why did the authors present Figures 8-10? These do not provide any meaning without simulation results based on the properties achieved in this work.
4. This reviwer strongly suggests the authors to survey and cite more references related to squeeze mode or tensile /compressive mode of MRF.
Reviewer 3 Report
The authors have addressed my comments properly.
Author Response
Thank you for your comment very much.Reviewer 4 Report
The author responded appropriately to the reviewers' comments, and the manuscript was well revised to reflect the reviewers' comments.
Author Response
Thank you for your comment very much.